# Burn Them Right! Determining the Optimal Temperature for the Purification of Carbon Materials by Combustion

Emmanuel Picheau [1] , Ferdinand Hof [1], Alain Derré [1], Sara Amar [1] , Laure Noé [2], Marc Monthioux [2]
and Alain Pénicaud [1],*

[1]   Centre de Recherche Paul Pascal (CRPP), UMR5031 CNRS, Université de Bordeaux, 33600 Pessac, France;
     emmanuel.picheau@crpp.cnrs.fr (E.P.); ferdinand.hof@gmx.de (F.H.); alain.derre@crpp.cnrs.fr (A.D.);
     sara.amar@crpp.cnrs.fr (S.A.)
[2]   Centre d'Elaboration des Matériaux et d'Etudes Structurales (CEMES), UPR8011 CNRS,
     Université de Toulouse, CEDEX 04, 31055 Toulouse, France; noe@cemes.fr (L.N.);
     marc.monthioux@cemes.fr (M.M.)
*   Correspondence: alain.penicaud@crpp.cnrs.fr

**Abstract:** A new purification procedure for carbon nanoforms is proposed. It was tested on multiwall carbon nanotubes (MWCNTs) prepared by arc discharge, which is among the most challenging of cases due to the chemical and structural similarity between the MWCNTs and most of the impurities to be removed. Indeed, the various methods for synthesizing carbon nanoforms lead to a distribution of carbonaceous products, such as carbon shells, carbon spheres, fullerenes, and a variety of other species. Thus, many strategies to purify the desired products have been developed. Among the most successful ones, thermal oxidation (combustion) seems particularly efficient. To be successful while preserving a reasonable amount of MWCNTs, the combustion temperature has to be carefully selected. Moreover, the ideal combustion temperature does not only depend on the material to be treated but also on the overall system used to perform the reaction, including the reactor type and the parameters of the gaseous reactant. Typically, the optimization of the purification relies on multiple experiments and analysis of the products. However, to the best of our knowledge, a strategy to determine a priori the most suitable temperature has not been reported yet. We demonstrate here that a thermogravimetric method, namely the constant decomposition rate thermal analysis (CRTA), is particularly well adapted to answer this question. An isothermal treatment based on the results obtained from a CRTA program allowed arc-MWCNTs to be successfully purified from graphenic shells while optimizing the yield of the MWCNTs. This strategy is believed to be valuable not only for purifying MWCNTs but also for the purification of other carbonaceous forms, including new carbon nanoforms.

**Keywords:** carbon material purification; carbon materials; carbon combustion; constant decomposition rate thermal analysis; thermogravimetric analysis; carbon nanotube purification



## 1. Introduction

The bulk synthesis of carbon nanoforms often requires removing carbon byproducts. Determining the right purification procedure is challenging because of the similarity in chemical nature between the phase to retain and the phase to eliminate. The need arose in the 1990s, once multiwalled carbon nanotubes (MWCNTs) were found to be formed by the same catalyst-free electric arc-discharge process [1] used for the large-scale synthesis of fullerenes [2]. Indeed, although the material synthesized by the arc discharge is 100% carbon-made, only about 30 to 70% (weight) of the solid produced corresponds to actual MWCNTs. The rest are carbon nanoparticles such as amorphous carbon, graphenic (i.e., made of graphene) shells, and fullerene derivatives such as nanohorns [3,4]. Purifying the material while being highly selective was then a difficult task not only because of the chemical similarity but also the structural similarity between the unwanted material and

the MWCNTs, as they all are graphene-based carbons (but amorphous carbon, which is highly rich in $sp^3$-C). As early as 1993–1994, the need to increase the MWCNT yield was fulfilled by the purification of the powder by combustion i.e., oxidative thermal treatment or gasification of the unwanted carbon material [5–8]. Combustion has been reported using different types of gaseous oxidants such as oxygen [7], air [6,7], and $CO_2$ [5]. The selective intercalation of graphenic impurities with species able to catalyze the combustion has also been successfully proposed to increase the combustion selectivity [3,9]. Combustion has been shown to be very efficient in purifying as well as opening the tubes [3,5,6,8], a fact that has been exploited early for carbon nanotube filling [6,8,10,11]. Even though the CNT content reached can be very high, a disadvantage was that only a small fraction of the whole initial material, about 1–2 w%, remains at the end of the purification process. This is a consequence of the low selectivity of combustion, i.e., the small difference between the combustion rate of CNTs versus that of impurities [12]. Thus, being able to select the adequate temperature to favor the burning of impurities over CNTs would enable a major improvement [12]. However, to the best of our knowledge, a versatile strategy for the determination of the ideal temperature for the purification of multiphase carbon powders has never been discussed in studies reporting oxidative thermal purification. This question seems all the more important since, as is known for carbon combustion, this temperature not only depends on the carbonaceous material but may also depend on the oven used, as well as the oxidizing agent and the hydrodynamic conditions (flow profiles, flow rates, partial pressure, etc.) [13,14]. A change in one of those parameters may influence the combustion kinetics as well as both the optimized purification temperature and final material distribution, making an optimization required for every parameter set.

Thermogravimetric analysis (TGA) equipment is commonly used to easily study gas reactions with solid materials. In those, the user usually imposes a temperature profile to a material and records the weight loss versus temperature. At the beginning of the 1960s, F. and J. Paulik as well as F. and J. Rouquerol independently introduced a new TGA method, particularly well adapted for the determination of thermal events [15–18]. Many variations of those methods have been developed under different names, e.g., constant decomposition rate thermal analysis (CRTA) and stepwise isothermal or quasi-isotherm thermal analysis [19]. Unlike standard TGA procedures in which a temperature profile is decided prior to the measurement, during CRTA, the apparatus adapts the temperature to the material's weight loss during the measurement. This dynamic method is controlled by a feedback loop taking the weight-loss rate as a parameter to be kept constant. Unlike standard constant heating rate programs, the CRTA allows successive thermal events close in temperature (4 °C only here) to be perfectly discriminated, since the machine stops heating as soon as an event is detected [15–19].

In this paper, the performance of CRTA in purifying a complex mixture of multiphase carbon nanoforms with high selectivity and high yield is investigated, taking the example of arc-discharge MWCNTs.

## 2. Materials and Methods

### 2.1. Raw MWCNTs

The MWCNT powder used in this study was purchased from Sigma Aldrich (406074) and came from an electric arc discharge synthesis (catalyst-free process). The MWCNTs possessed distributions of outer diameters, lengths, and numbers of graphene layers in the ranges 7–12 nm, 0.5–10 μm, and 5–30, respectively (commercial datasheet). Figure 1 shows the transmission electron microscopy (TEM) images at different magnifications of the raw MWCNTs. For the specimen preparation, some TEM grids (copper, covered by either a lacey or a continuous formvar/carbon film) were rubbed onto the powder (the same method for all the TEM images of the paper, except for Figure 1c–e for which the powder was dispersed by sonication in ethanol before drop casting on top of the TEM grid). In Figure 1a, one can see that the carbonaceous material was present under the form of aggregates, which did not only contain CNTs but also other carbonaceous particles. In fact, the supplier datasheet

states that about 30 w% of the powder corresponds to CNTs, as expected for arc discharge synthesis [3]. Figure 1b shows a magnified image of one aggregate revealing the presence of CNTs protruding from it. Figure 1c allows distinguishing the presence of CNTs and other carbon nanoforms considered as impurities. Figure 1d and e focus on a CNT end and on an impurity, respectively. As expected from the synthesis process type, Figure 1d confirms that the CNTs were closed at their extremities and possessed a large number of walls (5 to 30 according to commercial datasheet). Figure 1e shows a graphenic particle, typical of the kind of impurities found in such powder, and there were a majority of them.

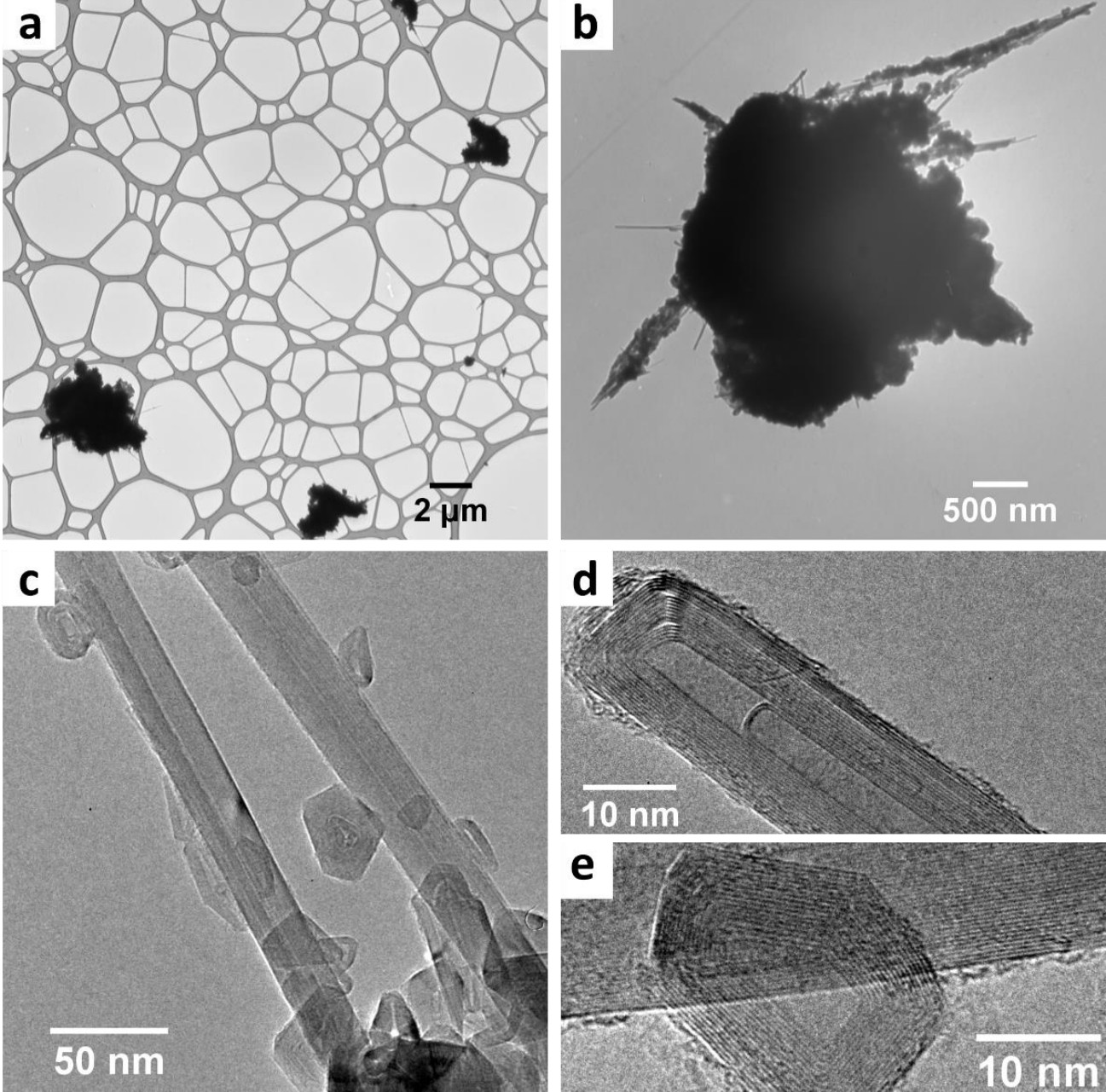

**Figure 1.** (**a**–**e**) TEM images of the raw material (before treatment) at different magnification scales. (**d**) Focus on a CNT end showing that CNTs are closed at their extremities. (**e**) Focus on a graphenic particle, typical of the kind of impurities present in the powder.

*2.2. CRTA Measurement*

First, 6.7 mg of the raw powder was placed in a platinum crucible previously cleaned with hydrochloric acid (37 %-sigma Aldrich), in order to perform the CRTA analysis. The analysis was conducted in a thermogravimetric analysis machine (TGA-TA instrument TAQ50) by two gas cylinders (Air Liquide): the synthetic air needed for the combustion and nitrogen to protect the balance. The balance flow ($N_2$, coming from above the crucible) was set at 40 mL/min, and the mass flow (synthetic air coming from one side) was set to 60 mL/min. The CRTA program used was the following:

1. Ramp 50 °C/min to 250 °C;
2. Abort next segment if %/min > 0.20 ("%/min" corresponds to the weight-loss rate);
3. Ramp 2.5 °C/min to 700 °C;
4. Abort next seg if %/min < 0.05;
5. Isothermal for 1000 min;
6. Repeat segment 2 until 700 °C

As we will see in the discussion section, the CRTA measurement (Figure 2) was used to choose the temperature treatment described hereafter. In fact, this method enabled detection of the presence of different materials by a direct reading of steps on the temperature profile (instead of slope changes on the weight-loss curve, which is unprecise).

*2.3. Isothermal Treatment*

The treatment consisted in the combustion of 70 w% of the powder, at 590 °C, in the same TGA equipment used for the CRTA analysis and under the same gas flow conditions. The remaining material was collected for analysis (TEM and Raman spectroscopy) and is referred to as the "treated material" in the following.

*2.4. TEM/Raman Acquisition and Analysis*

Figure 1c–e were obtained with a Philips CM30 ($LaB_6$ electron source) operated at 150 kV. Figures 1a,b and 3a,b were obtained on a Hitachi H600 operated at 75 kV. The high resolution (HR) TEM (Figure 3c,d) were obtained on a FEI Tecnai F20 SACTEM (Cs corrected) operated at 100 kV.

For the Raman analysis, a few milligrams of the powders (raw and treated materials) were flattened on a double-face scotch-tape deposited on a glass slide. Micro-Raman measurements were performed on an XploRA spectrometer (Horiba) at 532 nm (2.33 eV) excitation wavelength (laser spot size ~1 μm), calibrated on the HOPG G-band. Spectra were acquired with a cooled Andor CCD detector, using a 1200 lines per mm grating and a 4-s acquisition time, while the laser was filtered at 10 % of its maximal power (of ~14 mW). A mapping of $32 \times 32$ spectra on a surface of $180 \times 180$ μm$^2$ was recorded. The G and D characteristic bands of graphenic carbon materials were fitted using LabSpec6 software using Lorentzian functions. The data set obtained was analyzed with Origin 9.2.

## 3. Results

*3.1. CRTA Analysis*

Figure 2 shows the CRTA analysis. Both the weight loss (blue curve) and the temperature (green curve) are reported versus time. First, as the material did not contain any catalyst, the weight-loss curve showed no residue. The second fact to notice is the change in the weight-loss slope after around 70% of weight loss (after ~350 min). Remarkably, after burning 70% of the material at an almost constant temperature (see the green curve), the burning rate decreased and was stabilized at a temperature only a few degrees above that needed to burn the first 70%. Beyond that point, the apparatus needed to constantly increase the temperature to maintain a constant weight loss. These observations show that at least two kinds of carbon structures were burning with different kinetics. The structural difference highlighted here corresponds to the curved and the defective zones on one side and the straight graphenic parts on the other side. Both structural types were present on the

CNTs and on the graphenic particles but in different proportions. Since graphenic particles possess more curved parts than the CNTs (and were mostly localized at the CNT tips) the first 350 min of the measurement should have left the CNT walls almost unreacted and burned away a majority of the graphenic particles and tube tips.

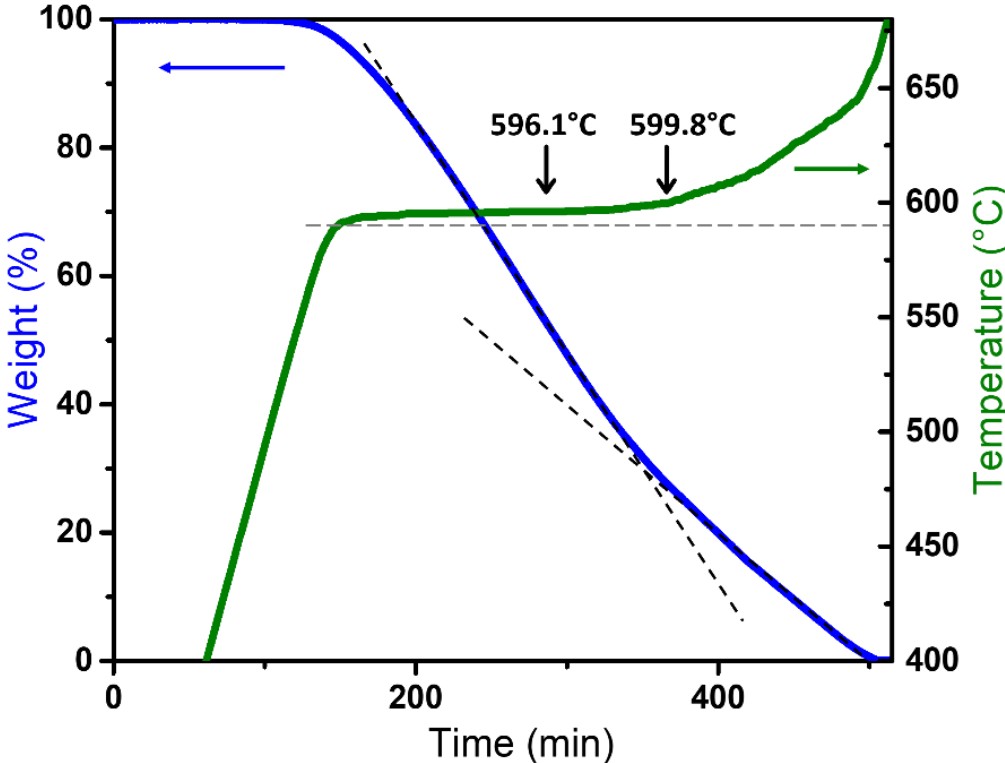

**Figure 2.** Thermogravimetric curve corresponding to the CRTA measurement of the raw material. The blue curve shows the % weight loss of the material versus time. The green curve represents the resulting temperature adopted by the apparatus (based on the weight-loss rate) versus time. The horizontal grey dotted line represents 590 °C.

### 3.2. Treated Material

### 3.2.1. TEM

To test the hypothesis proposed at the end of the previous section, we recovered the remaining material after the treatment described in Section 2.3 (isothermal treatment at 590 °C until 70% weight loss). Figure 3 shows the TEM and HRTEM images of the treated material (the 30 w% remaining material). At the microscopic scale, it is possible to see in Figure 3a that contrary to the raw material, the treated material did not present agglomerates; instead, the MWCNTs were spread across the image, and the impurities represented probably no more than a few percent. It is in fact possible to see in Figure 3b that almost all the graphenic particles were burned away, even inside a CNT intertwining. It was finally observed that many CNTs were opened as shown in Figure 3c,d. The observation that almost only the straight graphenic part of the MWCNTs (the "walls") stand after the treatment validates the abovementioned hypothesis.

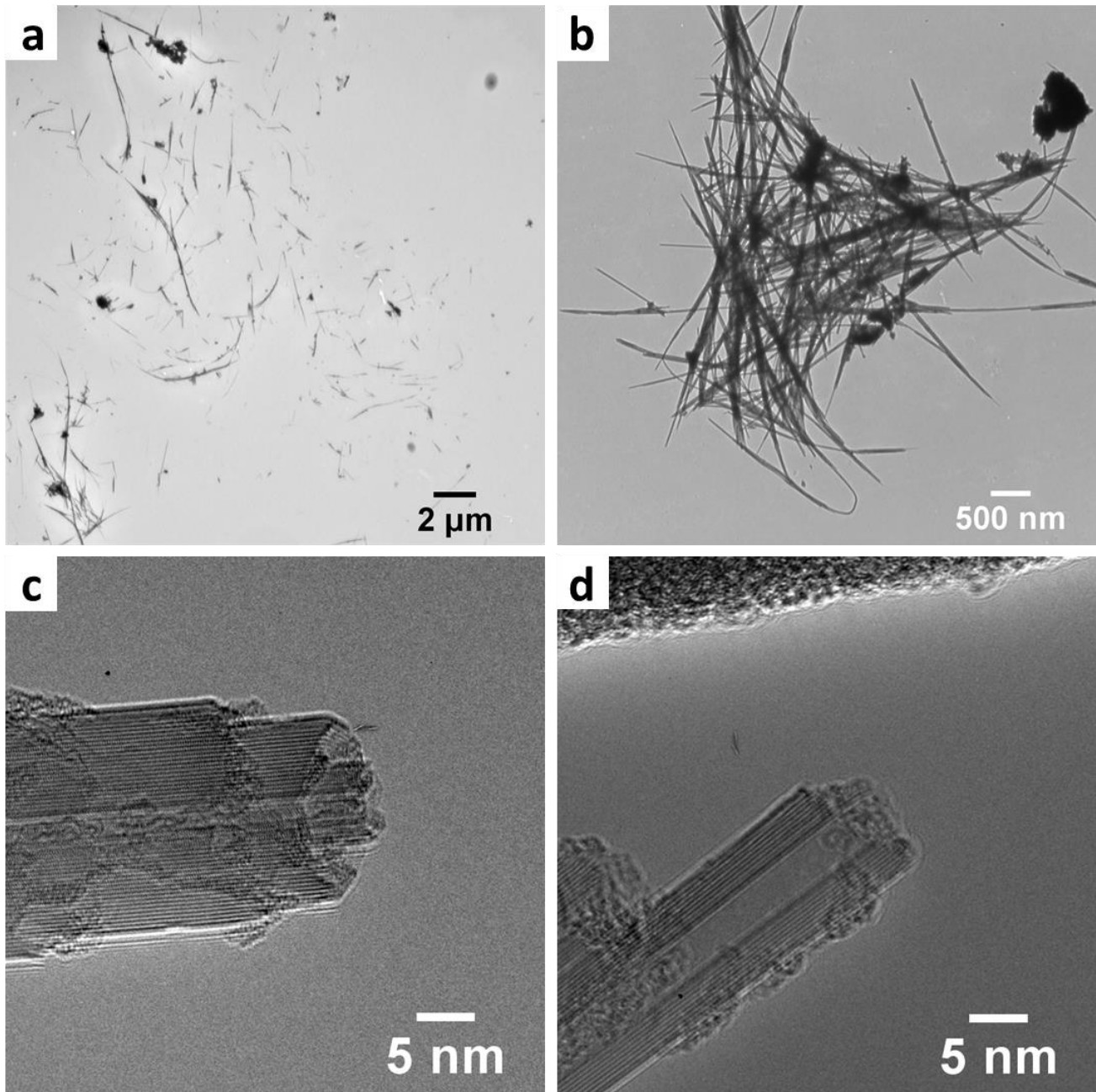

**Figure 3.** Transmission electron microscopy images of the material after treatment). (**a**,**b**) Low magnification. (**c**,**d**) High resolution images. A significant amount of CNTs have been opened (see text).

### 3.2.2. Raman Analysis

Figure 4 shows a Raman statistical analysis of the powder before and after isothermal treatment, based on 1024 individual spectra. Figure 4a,b report the ratio of the area of the D and G peaks ($A_D/A_G$) and the full width at half maximum for the G band ($FWHM_G$), respectively. Both depend on the presence of defects (any symmetry breaking of the honeycomb lattice such as local $sp^3$-C point, line defects, grain boundaries, Dienes defects [20], curvature changes, etc.), and both increase with the defect density [21–26].

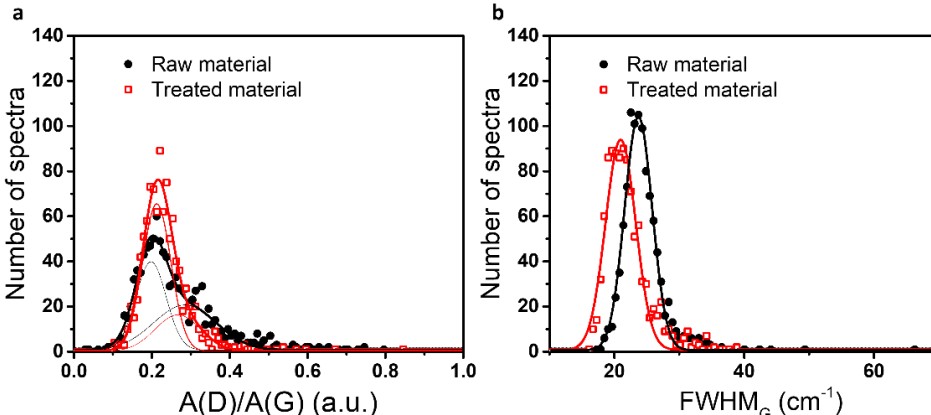

**Figure 4.** Comparisons of Raman data for the raw (black dots) and treated (empty red squares) materials, built on 1024 individual spectra. (**a**) Distribution of the area ratio of the Raman D band over the Raman G band. This distribution is based on a histogram with bin size: $8.33 \times 10^{-3}$ a.u.). The thin lines are Gaussian functions used for the fitting of the curves (thick lines). (**b**) Distribution of the full width at half maximum (FWHM) for the Raman G band and Gaussian fits. This distribution is based on a histogram with bin size: $0.58 \ \text{cm}^{-1}$.

Interestingly, the $A_D/A_G$ distribution (Figure 4a) needs two Gaussian functions to be fitted. This suggests the presence of at least two kinds of materials. Comparing the untreated versus treated materials, the area ratio of the Gaussians dropped from ~1.1 to ~0.4, suggesting that at least one kind of material, the most defective one, was burned away by the treatment (purification). The Raman resonance effects being a priori different for the different materials, no quantitative analysis can be made from the measurement. In Figure 4b, one can see that the treatment has narrowed the G band from a mean FWHM of 23.7 to 21.0 $\text{cm}^{-1}$. This is fully consistent with a purification of the material [21]. Comparing the mean Raman spectra before and after treatment (see Figure S1), almost no difference is actually visible, and it is quite remarkable that statistical analysis highlights the difference between those two datasets [27].

## 4. Discussion

In our example, the CRTA allowed us to determine that the reaction started at around 590 °C, and that the first thermal event burned away around 70% of the material. Those two values were used for the isothermal treatment. However, a consequence of the close structural and chemical similarity between the graphenic shells and the MWCNTs is that the latter were inevitably opened at their ends. However, in this specific case, the result can be seen otherwise: using CRTA allowed opening the MWCNTs at an optimized temperature, removing the unwanted carbon shells while burning a minimum of the CNT walls. Although neither the opening yield nor the purification were quantified, the purification yield is estimated as high on the basis of the TEM observations and statistical Raman analysis. It should be noted that the opening of the tubes in the treated material necessarily involves an increase in the Raman D band and a broadening of the G band, which seems to have been compensated for by the carbonaceous-particle elimination. As a result, by simply selecting the adequate temperature, the treatment enabled recovery of as much as 30% of the initial powder containing a high MWCNT fraction; as MWCNTs represented only about 30 w% of the initial material, this shows a rough estimate of purified yield close to 100%.

The CRTA shows that the combustion of impurities (i.e., graphenic particles and tube ends, as they both contain similar topological defects, which are the preferential sites for the oxidation to start) occurs before that of the tube walls, whereas these events cannot be distinguished in the more classical approach of a constant heating rate measurement (see Figure S2).

### 4.1. A Defect Based Selectivity

The rationale for combustion to purify and open MWCNTs comes from the small reactivity difference of MWCNT walls versus impurities and tube ends [12]. Indeed, the carbon combustion reaction does not occur on random sites but on specific ones called the active sites, which all together form the active surface area (ASA) [28]. The ASA corresponds to the portion of the total surface area (all carbon atoms at the surface of the material) able of oxygen chemisorption. The ASA includes dangling bonds, strong curvature sites, vacancies, and topological defects, such as carbon pentagons and Dienes defects. While all sites of the ASA can chemisorb oxygen, for a given temperature, only a fraction of them participate as reactive intermediates allowing the desorption of $CO/CO_2$, leaving a reactive site behind. This last fraction is called the reactive surface area (RSA). High curvature and strained band angles in tube ends and impurities make them more reactive than the hexagons that compose the CNT and shell walls, as also shown by calculations in [29]. If only the most reactive part of the ASA is thermally activated into RSA, the combustion will progress on the carbon material in the vicinity of the carbon desorption into $CO/CO_2$ since a reactive site is formed by the product desorption (this explains the telescopic aspect of oxidized MWCNTs often observed and visible in Figure 3c). Under such circumstances, MWCNT walls can stand combustion longer than graphenic particles, because the distance between two reactive sites is longer on the walls of MWCNTs than on the graphenic particles.

### 4.2. Importance of the Temperature and Use of CRTA

If the temperature is kept close to the one at which the initial defects start to burn, the difference between the combustion of the curved and straight parts can be quite different, allowing a good separation of them (as in Figure 2). Increasing the collision probability between oxygen molecules and the RSA formed on the wall ends by $CO/CO_2$ desorption would have the effect of increasing the wall burning rate, decreasing the separation. As a result, to selectively start and keep the reaction on impurities and at the tube ends much faster that on the tube walls, thermal energy should be carefully selected: high enough for the reaction to occur on the impurity active sites but not too high in order to avoid damaging the tube walls, as far as possible. The key question for any purification process of carbon nanomaterials based on combustion is then: what is the ideal temperature for the oxidative treatment? Up to now, this implied that, prior to treatment, a study to evaluate the most adapted temperature in an overall system, including oven technology and geometry, gaseous parameters, etc., had to be conducted. The standard strategy to conduct this was to perform several experiments at different more or less arbitrary temperatures and to analyze the remaining powder. To be able to discriminate two events thermally distant by only 4 °C, as in our example, was thus very unlikely with such a strategy. In CRTA, when a certain weight loss is detected (first thermal event), the first segment stops, and the apparatus program keeps the temperature constant (second segment) until the material weight loss diminishes below a user-defined threshold (see line 4 of the program in Section 2.2). Then, the controller starts to heat again to find the next thermal event, in a loop. This may allow the controller to self-find the temperature for the combustion of different species inside the powder. As a result, with a good choice of parameters for the CRTA program (that may need adjustment), the temperature at which the impurities start to burn can be precisely determined. Several types of impurities with distinct structural or chemical features would be even more easily discriminated than in the challenging example used here. If such impurities had been present here, more steps would be visible on the temperature profile, at lower temperatures, without necessarily implying a difference in the weight-loss rate.

### 4.3. Role of Diffusion

One central aspect of combustion is the possibility for the reaction to occur under diffusion regimes [13,14,30,31]. As for all solid–gas reactions, the gaseous reactant has to reach the solid surface by passing through a boundary layer, by diffusion. A competition between the gaseous reactant feeding the surface by diffusion and its consumption by

the surface reaction takes place during the reaction. If feeding the reactant is faster than consuming it, then the reaction lays in what is called the kinetic regime. Otherwise the reaction lays in one of the diffusive regimes as described by Walker and coworkers in reference [13]. The reaction selectivity discussed in this paper only stands if the reaction is in the kinetic regime. This may very well explain the success of Park and coworkers in the high yield purification of MWCNTs by combustion inside a rotating furnace [29]. In fact, stirring the material during the combustion reduces the diffusion limitations versus an immobile powder. While it has been shown over and over that the carbon combustion can easily suffer diffusional limitation [14], the use of "low" temperature, i.e., temperature close to the ignition temperature, favors the kinetic regime [13,31]. In the CRTA program, the temperature at which the apparatus will stabilize first (first isothermal step), depends on the choice of the criterion for the weight-loss rate. This criterion should consequently be selected to allow the reaction to be slow enough compared to the oxidant feeding, in order for the reaction to occur in the kinetic regime. Finally, the use of the CRTA program in equipment allowing the homogenization of the powder by stirring the powder during the measurement could probably allow reaching a higher selectivity for the carbon combustion, e.g., to dissociate between different kinds of impurities or even CNTs with different diameters.

## 5. Conclusions

The CRTA, an uncommon TGA program in which the material weight loss controls the temperature of the machine, was successfully employed to determine the good parameters for arc-discharged MWCNTs purification. Thanks to a good selection of the temperature which also avoided diffusion limitations, combustion was selectively performed on the curved part of the carbon materials, as far as possible. This allowed collection of about 100% of the starting MWCNTs (initially present at ~30%), mostly opened at their end and in a highly pure fraction. Beyond the purification of arc-discharge MWCNTs, the use of the CRTA program may prove very valuable for the purification of any kind of carbon material including those yet to be discovered. While here the method was demonstrated in the absence of a catalyst, the same strategy could also be successful in principle in the presence of a catalyst, which is the case of raw single-wall carbon nanotubes, although this would merit further investigations.

**Supplementary Materials:** The following are available online at https://www.mdpi.com/article/10.3390/c8020031/s1, Figure S1: Normalized mean Raman spectra for the D and G region of raw and treated materials. Figure S2: Thermogravimetric curve of the raw material in the classical approach of constant heating rate measurement (heating rate: 2.5 °C/min).

**Author Contributions:** Conceptualization E.P.; methodology E.P., F.H. and M.M.; formal analysis E.P., F.H. and M.M.; investigation E.P., F.H., S.A. and L.N.; resources A.D., A.P. and M.M.; data curation E.P.; writing—original draft preparation E.P.; writing—review and editing E.P., F.H., M.M. and A.P.; supervision M.M. and A.P.; project administration M.M. and A.P.; funding acquisition F.H., M.M. and A.P. All authors have read and agreed to the published version of the manuscript.

**Funding:** This research was funded by CNRS ANR Project ANR-16-CE24-0008-01 "EdgeFiller", Individual research fellowship HO5903/1-1 from Deutsche Forschungsgemeinschaft (DFG), and CNRS (Momentum grant of FH).

**Institutional Review Board Statement:** Not Applicable.

**Informed Consent Statement:** Not Applicable.

**Data Availability Statement:** The data presented in this study are available in the article.

**Conflicts of Interest:** The authors declare no conflict of interest. The funders had no role in the design of the study; in the collection, analyses, or interpretation of data; in the writing of the manuscript, or in the decision to publish the results.

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
