# Peer review of "Burn Them Right! Determining the Optimal Temperature for the Purification of Carbon Materials by Combustion"

_carbon_

Round 1
Reviewer 1 Report
In this article, the authors proposed a purification procedure for carbon nanoforms. A complex mixture of multiphase carbon nanoforms was purified by the constant decomposition rate thermal analysis (CRTA). However, generally, this manuscript is not innovative enough. Therefore, I would not recommend this manuscript for publication this time. Some problems are listed as follows:
- The purification procedure proposed in this work is just a simple combustion event, not novel enough. What’s the meaning of this work?
- Authors claimed that “These observations show that at least two kinds of carbon structures are burning with different kinetics. The structural difference highlighted here correspond to the curved and the defective zones on one side, and the straight graphenic parts on the other side.” But, the inflection point of the weight loss curve in Figure 2 is not clear enough, which is not conducive to the detection of intermediate products. So it is doubtful.
- In Figure 1c, the thickness of the wall of the MWCNT is about 25 nm, while the thickness of the material after treatment is about 5 nm in Figure 3d, please give a rational explanation.
- The opening yield and the purification of the MWCNThave not been quantified after CRTA in this work. It doesn't make a big sense for practical applications.
- The “Error! Reference source not found.” appears several times in the article, what does it mean?
Author Response
Please find attached our answers.

Reviewer 2 Report
The manuscript entitled "Burn them right ! Determining the optimal temperature for the purification of carbon materials by combustion" (MS#: carbon-1668718) is an original research article focused on the exploration of multi-walled carbon nanotube (MWCNT) purification based on an uncommon thermogravimetric analysis (TGA) routine which allows the discrimination of fine events.
It is fundamentally as well as practically important to better understand the thermal behavior of as-produced nanocarbons. To this end, the authors demonstrate the utilization of a feedback-loop-enabled dynamic measurement, constant (decomposition) rate thermal analysis (CRTA), for delivering a heat treatment protocol for MWCNTs purification. The manuscript is elegantly written, and the authors report pretty exciting results with in-depth discussions. Particularly, the cross-validation of Raman spectroscopy and TGA-CRTA is highly interesting. However, the manuscript is not free of flaws as outlined below under MAJOR ISSUES and MINOR POINTS:
MAJOR ISSUES
1) ON MOTIVATION/RELEVANCE
- a) It is not fully clear why the authors focus on MWCNTs. MWCNTs are relatively cheap products, and they are good to go in as-produced or after a brief thermal and/or minimal chemical treatment for several applications. Can the authors also specify which kinds of applications require such finely purified MWCNTs?
- b) Is it more relevant to explore single-wall carbon nanotubes (SWCNTs), or is it possible to do so in the first place? (The authors hint at this vaguely. But it would be helpful to the generalizability of this approach.)
- c) The thermal treatment proposed by the authors also eat up the tube caps. Can the authors the potential impact of this fact for applicability of their approach?
2) ON EXPERIMENTAL DESIGN
To better highlight the strength of CRTA in discriminating the two thermal events, it would be helpful to include a conventional TGA thermogram in this manuscript.
3) ON RESULTS & DISCUSSION
- a) In my opinion, it is also possible to fit two curves to the A(D)/A(G) ratio of the treated material (Figure 4).
- b) The authors spell out the most natural question to come in their Discussions section as follows: "It should be noted that the opening of the tubes in the treated material necessarily involves an increase of the Raman D band and a broadening of the G band which seems to have been compensated by the carbonaceous-particle elimination." But I think the authors should report some spectra or average spectra of neat and treated samples (after careful baselining and normalizing).
MINOR POINTS
1) ON WRITING FLAWS
The manuscript is elegantly written. Yet, the authors might want to check/amend with the following:
- a) "... temperature (4 degrees here! ) ..." --> ???
- b) "... Error! Reference source not found. ..."--> All figures have this problem.
- c) "... an important number of walls ..."--> ???
- d) "... gas bottles ..."--> "... gas cylinders ..."
- e) "... 32x32 spectra ..."--> "... 32´32 spectra ..." (multiplication sign)
- f) "... 0.58 cm-1 ..."--> "... 0.58 cm-1 ..." (minus sign)
P.S.: Format-related problems are repeated in a few instances.
2) ON VISUAL ITEMS
- a) I recommend the authors add a dotted horizontal line at 590 °C in Figure 2? (A thin and gray line would help to follow.)
- b) The font sizes are highly inconsistent in figures.
EXTRA POINTS:
1) Although it is a rather loose terminology, I think the term "graphenic" is a bit misleading in this manuscript as being used in two different contexts/dexcriptions: (i) Graphenic particles, and (ii) outer walls of MWCNTs. (To help differentiate, the addition of another loose terminology might help: "graphene-like.")
2) The Discussion section is unreasonably long and involves some elements that would normally fit in the Results or Conclusions sections. So, it would be helpful to split that section into a few pieces.
FINAL REMARKS
Overall, this manuscript reports a systematic, novel, and potentially impactful work but there is some room for improvement. Thus, I SUPPORT the publication of this manuscript upon REVISIONS. I hope that this report will help the authors improve their work for publication.
Author Response
Please find attached our answer.

Reviewer 3 Report
This paper studied the optimal temperature for the purification of carbon materials by combustion. Overall, this is an interesting and useful study and can be accepted if the authors can address the following comments.
1) All references citations cannot be shown in this paper. This should be improved.
2) When the combustion was applied for the materials, original materials such as MWCNTs may be lost. How about this part?
3) The discussion part needs some arrangement. Conclusion should be given.
4) As shown in Figure 3, after treatment, there are still some other materials. More details can be added.
Author Response
Please find attached our answer.

Round 2
Reviewer 1 Report
Authors have responded all questions ans revised the manuscript according to reviewers' comments, it can be accepted for publication.